# Sublimation of terrestrial permafrost and the implications for ice-loss processes on Mars

Thomas A. Douglas [1] & Michael T. Mellon[2]

Sublimation of ice is rate-controlled by vapor transport away from its outer surface and may have generated landforms on Mars. In ice-cemented ground (permafrost), the lag of soil particles remaining after ice loss decreases subsequent sublimation. Varying soil-ice ratios lead to differential lag development. Here we report 52 years of sublimation measurements from a permafrost tunnel near Fairbanks, Alaska, and constrain models of sublimation, diffusion through porous soil, and lag formation. We derive the first long-term in situ effective diffusion coefficient of ice-free loess, a Mars analog soil, of $9.05 \times 10^{-6} \, \mathrm{m^2 \, s^{-1}}$, ~5× larger than past theoretical studies. Exposed ice-wedge sublimation proceeds ~4× faster than predicted from analogy to heat loss by buoyant convection, a theory frequently employed in Mars studies. Our results can be used to map near-surface ice-content differences, identify surface processes controlling landform formation and morphology, and identify target landing sites for human exploration of Mars.

[1] U.S. Army Cold Regions Research & Engineering Laboratory, 9th Avenue, Building 4070 Fort Wainwright, Alaska, USA. [2] Cornell University, Cornell Center for Astrophysics and Planetary Science, 442 Space Science Bldg, Ithaca, NY, USA. Correspondence and requests for materials should be addressed to T.A.D. (email: thomas.a.douglas@usace.army.mil)

Mars once exhibited an energetic hydrologic cycle but permafrost and glaciation have dominated for the last billion years[1]. A range of dramatic landforms are thought to have developed from or been modified by ice sublimation and lag formation (Fig. 1)[2–6]. For example, sublimation tills have been found to protect glacial deposits from ice loss[2] while scalloped and dissected mantle terrains, and thermokarst-like pits and scarps, all may have formed by subsurface ice sublimation[3,7]. Complex valley terrains resembling the folded curvilinear ridge-and-trough texture of brain coral may have formed from a combination of ice flow and distortion and the differential sublimation of glacial ice beneath patterned soils[2,8]. High centered polygonal patterns may have resulted from differential sublimation of high-gravimetric- content ice beneath polygon troughs in contrast to polygon centers[9,10]. In addition, sublimation tills may coat or hide residual snowpack, glacial ice, frozen ancient lakes, or ocean deposits[7]. These features may indicate the presence of persistent water ice and, as such, they provide potential target landing sites for human exploration.

On Earth, meteorologic conditions favorable for sublimation are sensitive to air temperature and humidity and are overshadowed by snowfall and melt events, so few locations provide an opportunity for long-term (i.e., decades) sublimation measurements[5,11–13]. Some areas of the Dry Valleys of Antarctica are the only places on Earth where sublimation is believed to be a dominant natural geomorphological process[5,12]. However, long-term sublimation measurements are challenging due to changing meteorological conditions[13] and have thus never been made on Earth (or on Mars) limiting our ability to characterize, model, or predict landscape changes by sublimation. Sublimation-based processes are fundamental to geologic evolution elsewhere in the solar system, particularly on Mars[2,14], emphasizing the importance of Earth-analog studies.

To better understand how sublimation can develop landscape features in permafrost terrains on Mars we measured sublimation along the walls of the U.S. Army Cold Regions Research and Engineering Laboratory's Permafrost Tunnel near Fairbanks, Alaska (hereafter referred to as the Tunnel). The Tunnel is a 250 m long underground research facility excavated through

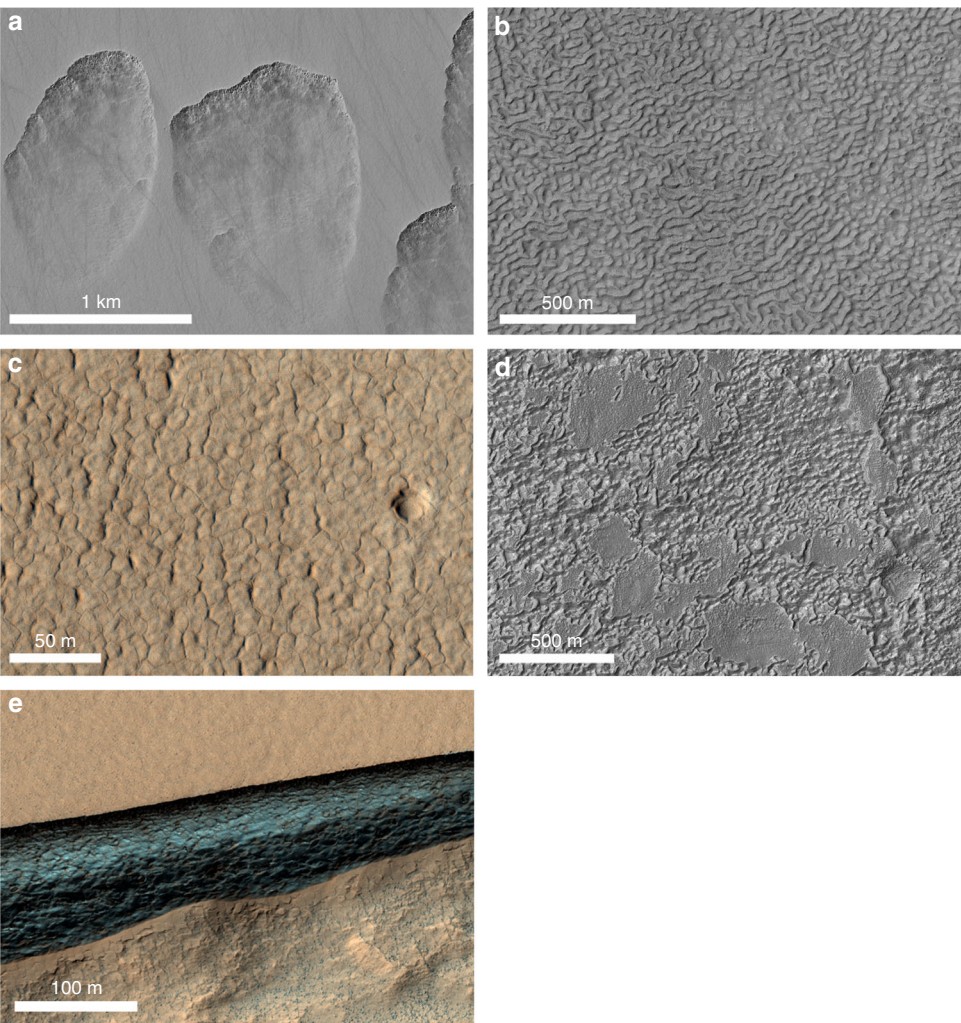

**Fig. 1** Images of potential sublimation landforms on Mars. **a** Scallop terrain—kilometer-scale pits with pole-facing back-wasting scarps thought to be due to ice ablation (HiRISE Image ESP_013952_1225); **b** brain coral terrain—riffle-like curvilinear ridge-and-trough textures usually found at the margins of glacial flow features (HiRISE Image ESP_023649_1360); **c** polygonal patterned ground exhibiting high-relief thought to be due to ice wedge loss through sublimation and differential erosion (HiRISE Image PSP_008896_2250); **d** dissected mantle terrain—hummocky inter-plateau deposits believed to result from loss of ice cement and subsequent surface subsidence (HiRISE Image ESP_049606_1390). In all images north is up and illumination is from the left; **e** ice-rich scarp—steep scarp exposing more than 100 m of massive ice where the scarp surface is maintained by active sublimation and mass wasting (HiRISE Image ESP_049461_1245)

syngenetic ice-rich permafrost since the mid-1960s and is unique on Earth[15]. No other tunnel provides similar access to a more than 50-year record of permafrost sublimation. The inside of the Tunnel provides a location with a stable air temperature ($-4.1 \pm 1.4\,°C$) and relative humidity ($91 \pm 3\%$) and no wind to remobilize sediment particles. Multiple periods of Tunnel excavation provide access to exposed surfaces where sublimation can be accurately quantified over a variety of exposure times. From these observations we are able to quantify water-vapor diffusion through undisturbed loess (a Mars analog soil[16]), and sublimation of clean ice by free convection.

## Results

**Measurements and observations.** We measured the rates of ice sublimation and ice-free lag formation at multiple locations within the Tunnel through two experimental campaigns. In the first campaign we repeatedly measured sublimation of freshly exposed ice and adjacent ice-cemented loess at one location over a 386 day period. For the second campaign we measured sublimation from these same two types of ice featured at various locations throughout the Tunnel representing exposure initiated at different times during the Tunnel's (then) 52-year record. From these measurements we constrain models of sublimation through soil pores and lag formation and derive the first long term in situ effective diffusion coefficient through ice-free loess, a Mars analog soil.

For the first experimental campaign we selected a location 32 m into the main Tunnel where an ice wedge and the surrounding ice-cemented loess are present at a shallow ($25°$ off of vertical tilting away from the tunnel floor) angle for which slumping and sediment detachments were not commonplace. This area was initially excavated in the mid-1960s. The ice-rich study feature is 1.5 m tall and 2 m wide (Fig. 2). We removed the roughly 10 cm thick covering of loess particles that remained following 52 years since excavation. The ice wedge and ice-cemented loess were gently scraped to a clean surface using wire brushes. We installed a series of rods into the permafrost. 10 cm deep holes were drilled into the ice wedge and ice-cemented loess at four locations using a drill with a 3 mm diameter bit. Stainless steel rods 3.2 mm in diameter were lightly tapped into the holes and reference marks were made on the rods where their exposure was even with the outer ice wedge surface. We similarly installed 12 cm long rods into the ice-cemented loess at four locations.

Repeat photographs and measurements of sublimation loss at multiple locations in the ice wedge and ice-cemented loess were collected for a year. Digital photographs were taken from a permanently mounted base each time the sublimation measurements were made (Fig. 2). The study site was not disturbed throughout the 386-day experiment. Initially, the ice-wedge feature was easily identified as dark ice in clear contrast to the ice-cemented sediment around it (Fig. 2). Over time, sublimation of the ice wedge was more rapid than the ice-cemented loess and the ice-wedge feature receded into the wall. This process continued until day 386 at which time the wedge ice had retreated 1 to 2 cm further back into the wall than the ice-cemented loess. To quantify ice-wedge ice sublimation the distance from etch marks on the installed rods (denoting the original freshly-cleaned ice surface as described above) and the current ice surface was measured repeatedly at four locations surrounding each rod over 386 days.

To measure desiccation of the ice-cemented loess and growth of the dry-loess lag repeated measurements of the depth of sublimation into the frozen sediment were made over 386 days using a 0.8 mm diameter steel pin with graduated markings. To make a measurement, the pin was connected to each of the rods

with a short wire and sublimation depth was made above, below, left, and right of the rod at the same time that the ice wedge measurements were made.

At the conclusion of this 386-day experiment we extracted twelve 15 cm long by 7 cm diameter SIPRE cores from the ice-cemented loess adjacent to the ice wedge to measure the gravimetric moisture content and soil composition[17]. Additional cores were collected from a variety of locations representing ice-cemented loess throughout the Tunnel ($n = 28$). Values ranged from 0.6 to 2.1 g of water (ice) per gram of dried loess. Based on an intrinsic density of $2.56\,g\,cc^{-1}$, these values translate to 63 to 85% ice by volume, at the low end consistent with high-porosity loess previously measured from the Tunnel[18] and at the high end consistent with the presence of ice lenses (see also Fig. 3c). Forty additional SIPRE core samples of wedge ice were collected at this location ($n = 10$) and other locations throughout the Tunnel ($n = 30$). The gravimetric moisture content of the wedge ice ranged from 250 to 810 g of water per gram of dried loess, less than 0.4% wt. soil content. There was no statistically significant difference between the gravimetric moisture contents at the 386 day measurement location and the remainder of the Tunnel for either the wedge ice or the loess (analysis of variance, $\alpha = 0.05$ in each case).

For the second experimental campaign we examined the long-term development of ice-free dry-loess lag and ice-wedge ice retreat at locations throughout the Tunnel. Several sections of the wall had been cleared of dry-loess lag at different times in the Tunnel's history for a variety of small experiments and excavations 10, 11, and 18 years ago, exposing fresh ice-cemented loess surfaces to sublimation (Fig. 3). Two major excavations in 2011 and 2013 provided an additional 75 m of new Tunnel exposure. In total, these locations allow access to 11 unique measurements of lag thickness over ice-cemented loess and three unique measurements of exposed wedge-ice sublimation representing exposure times ranging from 2.25 to 52.5 years. To accomplish these observations, we measured the horizontal distance from the flat outer plane of each ice wedge feature to the outer surface of the nearest wall at these multiple Tunnel locations representing the different times since excavation. Figure 3b includes a photo representing the retreat of ice wedge ice (relative to ice-cemented loess) following 5.8 years since excavation.

In 2015 we made lag thickness measurements overlying ice-cemented loess at multiple locations throughout the Tunnel where excavation occurred along main wall sections at known times of Tunnel excavation. These sites represent elapsed times since 1964, 1998, 2011, and 2013. A graduated steel rod was pushed into the sediment to refusal and the depth of the rod was noted. Fifty of these measurements were made from each of the different aged walls representing the four time periods since excavation. Additionally, we repeated these four measurement sets in 2016 after one year of additional lag was allowed to develop.

The Tunnel has a cooling system to ensure temperatures remain below freezing through the summer months. An array of thermistors reporting real time temperature and relative humidity has been operated in the Tunnel since the mid-1960s. In 2008 a system of thermistors and automatically recording dataloggers were installed to record hourly measurements of air temperature and relative humidity. A mean air temperature of $-4.1\,°C$ (with a standard deviation, SD, of $1.4\,°C$) and relative humidity of 91% (SD of 3%) were recorded in the main tunnel between 2008 and 2016. These values are similar to the real time measurements made since the mid-1960s. The standard deviations in the datalogger values are due to real temporal changes in environmental conditions as the cooling system varied during seasonal changes or thermal experiments were performed in the Tunnel. Mean temperatures also vary by $+/-0.7\,°C$ between different locations

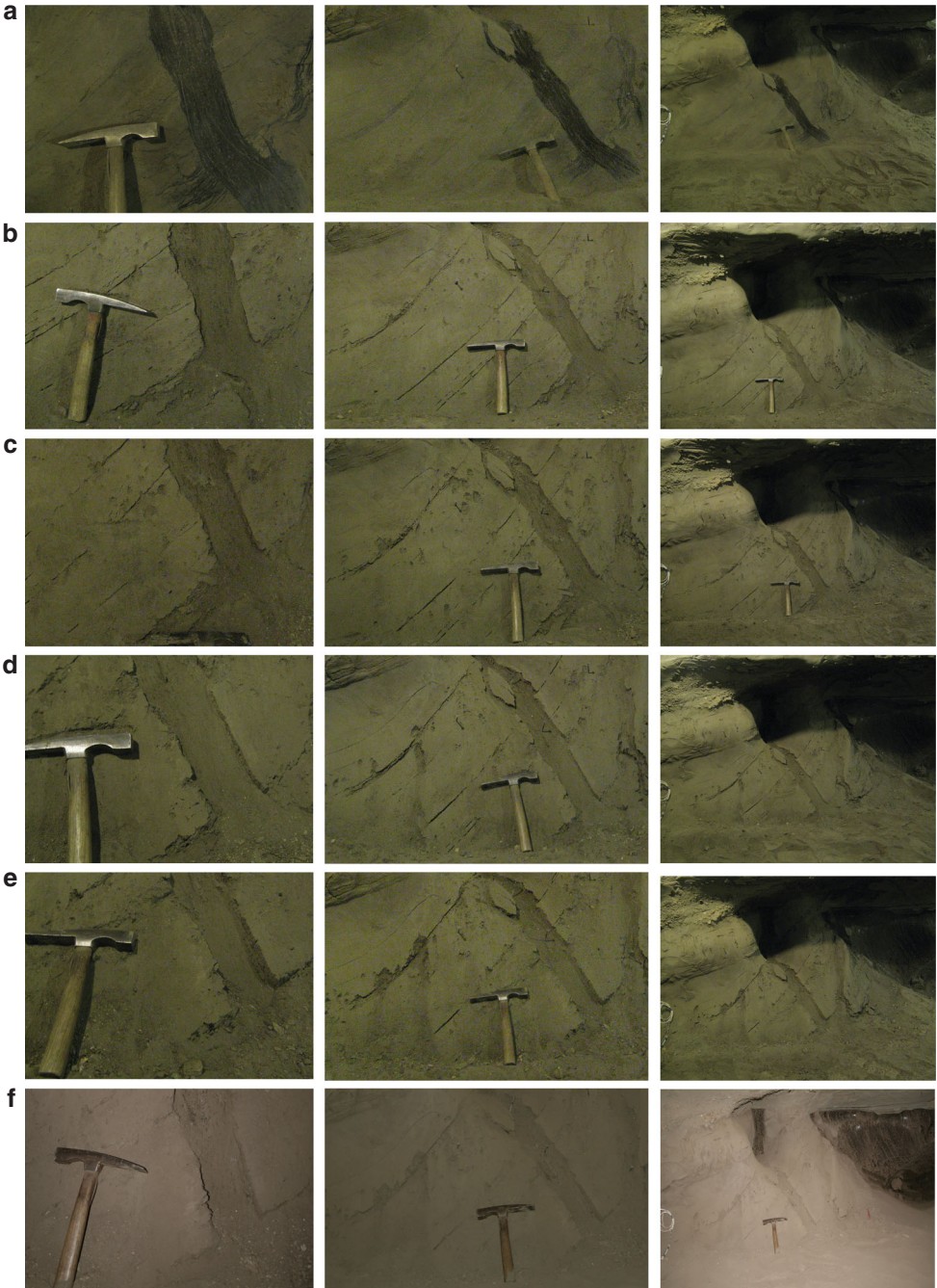

**Fig. 2** Repeat images of the ice wedge and ice cemented loess feature investigated over 386 days. Elapsed time, in days are **a** (0), **b** (64), **c** (163), **d** (184), **e** (309), and **f** (386). The ice wedge (dark colored vertical strip in the center of the images in the left column) and horizontal segregated ice (linear, thin, and dark colored) retreat into the wall as sublimation occurs more rapidly in the higher ice content material. Note the small silt, sand, and rock chip particles that start to fall away from the wall at day 163. The hammer, present in most of the photographs for a scale, is 32 cm tall. A larger ice wedge (dark black body to the far right of the right most column of photos) is visible but that was not instrumented as part of the small scale study

within the tunnel with a slight stratification of colder denser air near the floor. Deviations from the mean, typically for no more than 1–5 days, are associated with focused heating and cooling experiments in the Tunnel and they are not long lasting. Since 2008 the air temperature and relative-humidity deviations have ranged from 0 to −11.3 °C and 65% to 99%, respectively. Knowledge of the long-term temperature of the icy permafrost walls is limited (discussed below).

Since sublimation and vapor-diffusion processes are dependent largely on gradients in the water-vapor concentration, and since the relative humidity has an added dependence on temperature, we report the absolute humidity in terms of a water-vapor density. The average vapor density recorded between 2008 and 2016 was $3.2\,\mathrm{g\,m}^{-3}$ (SD of $0.4\,\mathrm{g\,m}^{-3}$). At different locations within the tunnel, the mean absolute humidity varied by about $+/-0.16\,\mathrm{g\,m}^{-3}$. During 2008–2009 we measured the temperatures and humidity profile from floor to ceiling (0–4 m) at the location of the 386-day experiment and found that air near the floor was 8% drier than air above 1.5 m, consistent with the buoyancy of humid air.

As shown in Fig. 4, we found that the rate of lag formation in ice-cemented loess over time follows a log normal fit (slope of

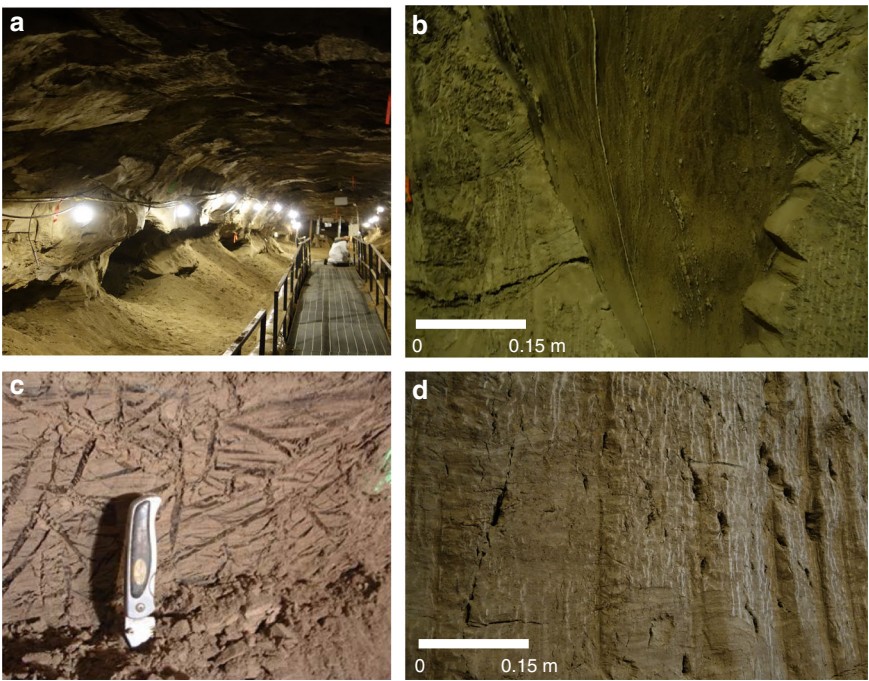

**Fig. 3** Photographs representing different surfaces present in the tunnel and their differential sublimation features. **a** The main tunnel showing the walls 52.5 years since excavation. Note the dark shadowy regions are where massive ice wedge features have retreated more than a meter into the wall. The roof and upper wall material represent pristine 52.5-year exposures. **b** Differential sublimation of ice wedge ice (dark color) and ice cemented silt (dark tan) in a wall exposed to sublimation for 5 years. **c** Reticulate-chaotic ice cryostructure with areas of ice (angular dark features retreated inward) in ice cemented silt. This exposure is three years old. **d** A 5 year old exposure exhibiting differential sublimation of segregated, lenticular-layered, and lens ice features

~1/2; $r^2 = 0.90$), and has resulted in nearly 10 cm of weakly cohesive dry-loess lag over 52.5 years. It is clear that in ice-cemented loess the rate of sublimation decreases over time as sublimation leaves an increasingly thicker protective dry-loess lag. In comparison, the rate of sublimation loss of wedge ice was 1.21 m over the 52-year record with a linear fit of the rate of ice loss with time ($r^2 = 0.98$).

**Vapor diffusion and lag formation.** Sublimation and desiccation of ice-cemented loess occurs by vapor diffusion through the porous-soil lag left behind as ice is lost and residual mineral and soil particles remain[19,20]. The sublimation rate decreases proportionally to the growing lag thickness (Fig. 4a), as governed by Fick's first law of diffusion, which relates the diffusive flux $F$ of a gas species to the vapor density gradient and the diffusion coefficient of the porous medium[21]. In this application, the diffusion of water vapor through the ice-free lag (assumed uniform) can be expressed as

$$F = -D\Delta N/Z \qquad (1)$$

where $\Delta N$ is the difference in the water-vapor density between the ice-cemented-loess/dry-loess interface and the Tunnel atmosphere across the lag thickness $Z$. $D$ is the effective diffusion coefficient of the dry lag, which depends on the pore structure via $D = D_{12}\varepsilon/\tau$, where $D_{12}$ is the binary diffusion coefficient of water vapor in air, $\varepsilon$ is the loess porosity, and $\tau$ is the tortuosity of the pore space[20]. $D_{12}$ is proportional to temperature and air pressure as $T^{3/2}P^{-1}$.

This flux of water being lost can also be expressed in terms of the lag growth rate $dZ/dt$, porosity, and ice density $\rho_i$,

$$F = -\rho_i\varepsilon dZ/dt \qquad (2)$$

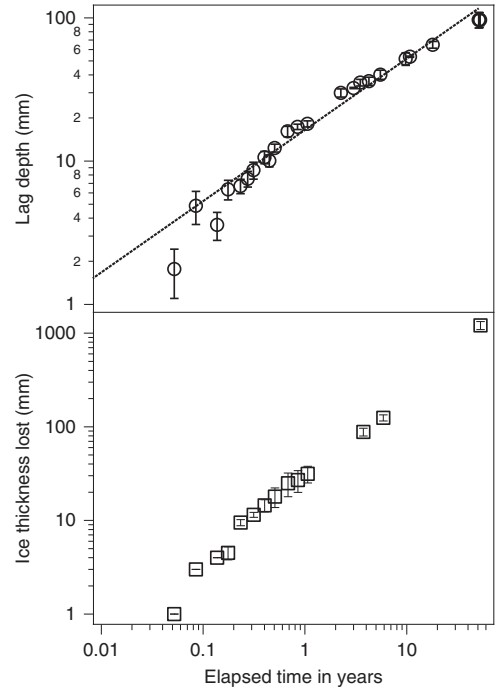

**Fig. 4** The 52.5-year record of sublimation measurements from the tunnel. Upper panel: ice cemented loess yields a log normal relationship with an $r^2$ of 0.90. This fit is denoted by the dashed line. Lower panel: loss of ice wedge ice over time. Vertical lines with horizontal bars represent plus and minus one standard deviation of the value for those measurements

Integrating Eqs. (1) and (2) yields,

$$Z(t) = \left[\frac{2D\Delta Nt}{\rho_i \varepsilon}\right]^{1/2} \qquad (3)$$

Equation (3) can be fit to the observed lag thickness over the Tunnel's 52-year history (Fig. 4) from which we obtain a best fit effective diffusion coefficient of $9.05 \times 10^{-6}$ m$^2$ s$^{-1}$ ($r^2 = 0.96$), representing the first determination of the diffusion coefficient in a terrestrial field setting.

In these calculations we assumed the wall temperature is isothermal with the measured average air temperature of $-4.1$ °C, consistent with reported subsurface temperatures in the area[22]. The temperature history of the wall-ice surface (the wedge-ice surface or the interface between dry-loess and ice-rich loess) has not been measured. We expect the bulk permafrost at the depth of the tunnel to reflect the local mean surface temperature over the past 30 to 100 years, estimated to be in the range of $-5$ to $-3$ °C[22–24]. However, the active cooling of air within the tunnel will slowly effect the wall surfaces such that the wall temperatures will gradually warm or cool toward the mean air temperature; large swings in air temperature will be greatly subdued by the large thermal mass of the permafrost, compounded by any insulating dry-loess layer; and latent heat of sublimation of wall ice will further cool the walls. As such $-4.1$ °C is a reasonable mean wall temperature, but may deviate slightly, the sensitivity of which is discussed below.

We also adopted an average absolute humidity of 3.2 g m$^{-3}$, and a lag porosity of 68% based on previous laboratory measurements[18]. In examining the overall lag growth over the 52.5-year history of the Tunnel we assumed constant environmental conditions. This assumption is supported by the linearity of exposed wedge ice loss (Fig. 4). In addition, we similarly examined the lag increase that occurred over one year (2015–2016) and obtained similar results. These data were collected at locations with a range of lag thicknesses, and corresponded to the same environmental conditions within the Tunnel during this single year.

**Sublimation of exposed ice.** In contrast to ice-rich loess, sublimation of exposed wedge ice, for which little entrained loess is available to form a protective lag, is driven by direct sublimation and free convection into a subsaturated atmosphere. This process has been modeled repeatedly in application to exposed ice on the surface of Mars using analogies to free convection heat loss[25–28]. In this standard model, dimensionless parameters for mass transfer are related by

$$\text{Sh} = C * (\text{Gr Sc})^m, \qquad (4)$$

where $C$ and $m$ are empirical values related to the specific geometry of the buoyant convection and to the Rayleigh number[29]. In Eq. (4), the Sherwood number Sh represents the ratio of convected mass transfer to molecular diffusion, the Grashof number Gr represents the ratio of buoyant force to viscous force, and the Schmitt number Sc represents the ratio of viscous diffusion to molecular diffusion, and are given by

$$\begin{aligned} \text{Sh} &= h_m L/D_{12}, \\ \text{Gr} &= g\beta(C_s - C_\infty)L^3/\nu^2, \\ \text{Sc} &= \nu/D_{12}. \end{aligned} \qquad (5)$$

In Eq. (5) $h_m$ is the mass-transfer coefficient, $L$ is the vertical scale of the ice wedge exposure, $g$ is the gravitational acceleration, $\beta$ is the air-density ($\rho$) change with respect to sublimation and given by $-(\Delta\rho/\Delta C)/\rho$, $\Delta C = C_s - C_\infty$ is the water vapor concentration gradient, and $\nu$ is the kinematic viscosity of air.

For a vertical wall and the environmental conditions in the Tunnel, $C = 0.59$ and $m = 1/4$, which are derived empirically[30]. Equation (4) is then solved for the mass-transfer coefficient $h_m$, which when multiplied by the absolute humidity gradient $\Delta N$ results in the sublimation flux $F_S$ of water from a vertical exposure of ice.

$$F_S = C\frac{D_{12}}{L}\Delta N\left(\frac{g\frac{\Delta\rho}{\rho}L^3}{D_{12}\,\nu}\right)^m \qquad (6)$$

A similar approach has been applied to Mars ice sublimation from horizontal surfaces where $m = 1/3$ and $C$ is 0.14 or 0.17[25–28] with similar quantitative results.

Over the 52-year record we observed that wedge ice sublimates at a mean rate of about 0.023 m yr$^{-1}$ (~21 kg m$^{-2}$ yr$^{-1}$) and is linear in time (Fig. 4b), indicating relatively constant environmental conditions over time. Using the observed air temperature and absolute humidity in the Tunnel ($-4.1$ °C and 3.2 g m$^{-3}$, respectively) combined with Eq. (4) we obtain a sublimation rate of 4.8 kg m$^{-2}$ yr$^{-1}$, about 4× smaller than we observed. This discrepancy between our observed sublimation rate and the modeled rate is not explained by possible variations in free parameters within the limits of our Tunnel observations.

If we consider a scenario where drier air near the floor is buoyantly convected up the face of the ice wedge (8% drier air would yield 2.97 g/m$^{-3}$ of water vapor) we then obtain a sublimation rate of 10.2 kg m$^{-2}$ yr$^{-1}$, still 2× smaller than our observation. However, this scenario is weakened by the fact that this rising air would encounter more humid conditions as it convects to a position above the floor. This would rapidly reduce any buoyancy advantage that might have resulted from a drier source. Therefore, the discrepancy between the standard model and our observation is closer to 4×.

## Discussion

We have derived the vapor-diffusion coefficient of the intact Tunnel loess to be $9.05 \times 10^{-6}$ m$^2$ s$^{-1}$. Previous measurements of collected (loose, structurally disturbed) samples of Tunnel loess[18] yielded an effective diffusion coefficient of $7.35 \times 10^{-6}$ m$^2$ s$^{-1}$, about 1.23× smaller than our in situ diffusion coefficient. This difference suggests that in situ cohesive loess retains a more openly-connected pore structure (e.g., lower tortuosity, $\tau$) than the disturbed loess samples with the same porosity. Such open structure may result from ice growth and soil freezing processes occurring when the wet loess initially froze within the pores 11,000–40,000 years ago. The process likely increased pore-to-pore connectivity, either through the expansion of water upon freezing within existing pores, or by grain-scale ice segregation cause by thin film migration along grain[31,32]. Such a grain-scale expanded structure could be retained after desiccation, supported by remaining grain contacts and mild cohesion.

The Antarctic Dry Valleys represent a rare natural permafrost setting on Earth where sublimation and vapor diffusion are dominant and liquid water is virtually absent[33]. Several studies of Dry Valley ground-ice stability and ice loss have used theory-based estimates of the effective diffusion coefficient of the ice-free permafrost till and report values of ~2–6.3 × 10$^{-6}$ m$^2$ s$^{-1}$ (refs. [11,13,29]). These Antarctic soils have generally lower porosity and tortuosity compared to Tunnel loess[18]. Accounting for these differences, our results suggest that an appropriate diffusion coefficient for Antarctic applications lies at the high end of this range, especially when examining permafrost desiccation and vapor diffusion through the desiccated lag (sublimation till).

The results from this study can also help to develop better predictive physical and conceptual models of the role of sublimation in the formation and evolution of geomorphologic

features in permafrost terrains on Mars. Our measurements provide a useful analog for various studies of Mars ice sublimation, permafrost desiccation, glacial-ice stability, and responses to climate changes. Fine-grained martian soils may exhibit a similar high-porosity structure to that of Tunnel loess after desiccation of interstitial ice. This analogy has been proposed as an explanation for a continental-scale deposit of low-thermal-inertia surface soil found in the southern high latitudes of Mars[34]. In this scenario, desiccation of ice-rich permafrost retains a high-porosity and low-thermal-conductivity structure analogous to Tunnel loess. In experiments conducted with sublimation of clay and ice mixtures, a high-porosity lag was produced and termed filamentary sublimation residue, which was proposed as an analog to sublimation lags formed on Mars and comets[35].

Based on the kinetic theory of gases, the binary diffusion coefficient scales as $T^{3/2}$ and $P^{-1}$, as noted above. On Mars, the mean temperature is −68 °C (205 K) and atmospheric pressure is 600 Pa, such that the diffusion coefficient for this same soil would be ~109× larger. However, the difference in diffusion through air (Earth atmosphere) and carbon dioxide (Mars atmosphere)[20] reduces this scaling by a factor of 1.65 to give a Mars effective diffusion coefficient of $5.99 \times 10^{-4}$ m$^2$ s$^{-1}$, ~66× larger. This value is similar to laboratory measurements conducted with glass spheres at Mars pressures[36]. This value is also 4 to 5 times larger than has been utilized previously in Mars theoretical studies[18–20,37,38], which can be mainly attributed to the high-porosity and low tortuosity structure of this analog to martian soil.

The larger diffusion coefficient we derive for Mars would yield higher rates of sublimation of unstable ground ice and rapid diffusive exchange of water between the martian regolith and atmosphere. Orbital cycles and resulting climate change on Mars have been previously examined[4,38], along with associated loss (or gain) of ground ice as stability conditions change[3]. Relict ice (unstable ice left over a past climate state) may persist if a sufficient delay of ice-loss follows the most recent period of stability. Rapid diffusion of sublimated water vapor decreases the potential for relict ice to occur. Sublimation till and karst landforms may form from the loss of a massive subsurface ice matrix, resulting in visible differential subsidence which may be linked to lag thickness and lag growth, and to contrasting ice-soil ratios[3,5,33]. Glacial ablation, thermokarst pits and scarps, high-relief patterned ground, and curvilinear ridge-and-trough textures resembling brain coral may all form rapidly in the current martian climate.

Sublimation of exposed ice on Mars has been examined for decades using approaches similar to Eq. (6). This approach was proposed to address the potential for martian surface ice to undergo melting vs sublimation when heated[25]. Subsequent studies have further explored water stability and melting potential[32], the lifetime of surface ice[23,28,39], and the stability of polar ice caps in relation to climate change[24,40]. Each of these studies have employed a variant of Eq. (6), assuming a horizontal ice surface, to examine buoyant free convection, sometimes in combination with forced convection resulting from martian winds[40,41]. In recent years, a number of laboratory studies have sought to validate Eq. (6) and its application to Mars studies. These measurements have focused on free convection sublimation of liquid water and ice, near or below freezing and at pressures from terrestrial ambient to low pressures comparable to[27,42–46]. Results from these studies have been inconsistent, pointing to both similar rates to those predicted by Eq. (6) and often to higher sublimation rates by as much as ×4. These highest rates are consistent with our in situ findings of wedge-ice sublimation within the Tunnel. Chittendon et al. (2008) suggested that 15 °C warmer air temperature in their experimental chamber could explain their 4× discrepancy between observations and Eq. (6). However, while warmer air can affect the buoyancy of humid air

plumes, the high thermal inertia of the ice body will prevent significant warming of the ice surface (especially under low pressure conditions) and thus cannot explain the discrepancy.

Furthermore, we similarly observed a 4× higher sublimation rate under nearly isothermal conditions within the Tunnel, further suggesting air temperature effects cannot explain their laboratory data. We conclude that Eq. (6), which is based on an analogy to heat loss by buoyant free convection, under predicts sublimation mass loss by buoyant free convection by about 4×. Application of this model to problems of sublimation on Earth and Mars should account for this discrepancy. On Mars, this difference is important when linking sublimation to the timing of naturally occurring secular or cyclic changes. For example: summertime sublimation from the martian polar caps, which is thought to be linked to the global atmospheric humidity[40]; the rate of evolution of non-polar ice exposures[7]; determining the concentration of soil in ice exposed by recent impacts[28]; and the lifetime and instability of transient liquid water[25–27].

In both cases of diffusion and exposed-ice sublimation a warmer than assumed wall temperature may help to explain some of these differences. Only a ¼ °C increase in the mean wall temperature would be needed for the diffusion coefficient to match the laboratory measurements of the disturbed loess. A 1 °C increase would be needed for observed sublimation to agree with theory. While these temperature increases are small, they are difficult to envision in light of cooler tunnel air temperature and the cooling effect of sublimation itself. What's more, no value of wall temperature provides a simultaneous match to both diffusion through loess and sublimation. Given the parameterized nature and heat-convection-analogy of the model of ice sublimation, a simpler explanation is that this model under predicts sublimation.

These measurements and analyses have direct application to Mars and Earth cryospheric processes. In polar desert environments such as the Antarctic Dry Valleys sublimation can dominate permafrost evolution. Similar conditions exist over all of Mars and studies of the climate and hydrological cycle extensively employ vapor diffusion in arid soils and sublimation of exposed ice. This improved understanding of the rates for ice sublimation, frozen soil desiccation, and lag formation enables a better assessment of water ice composition on Mars and supports the identification of opportune landing sites for future exploration. Water is essential for life on Mars. Therefore improving our understanding of the processes that control its distribution, concentration, and response to martian climate change is essential for ongoing exploration of the planet. From this terrestrial-analog study we can constrain, validate, and adjust our models as applied to Mars to better realize these planetary exploration goals.

## Methods

**Study site.** The Permafrost Tunnel is located 11 km north of Fairbanks in Fox, Alaska (64.9528 N, 147.6178 W). The main tunnel (adit), roughly 90 m long, was excavated horizontally into a gently sloping northwestward facing hill from 1963 to 1966. The Tunnel is 4 to 5 m tall and 5 to 6 m wide. Groove marks from the mining machine used in the 1960s excavation are still readily visible in many locations and are evidence of original wall and ceiling locations representing the 52-year record presented in this study. A gently sloping 50-m long ramp (winze) was excavated from 1968 to 1969. Unique experiments in 1999 and 2006 led to clearing of dry loess from multiple 10 m long by 2 m high exposures along the Tunnel walls. An additional 35 m of Tunnel was excavated in 2011 and 40 m more in 2013. All of these sections contain ice wedges and ice-cemented loess and the different excavation periods provide additional clean walls on which sublimation has acted since excavation. The temperature and relative humidity within the Tunnel have been maintained since the mid-1960s for Tunnel stability.

The Tunnel provides access to Late Pleistocene-aged aeolian loess, reworked loess, and alluvium and colluvium of the Goldstream Formation[23]. The Fairbanks area has a continental climate with a current mean annual temperature of −3.3 ° C[24]. Paleoclimate temperatures in this area are less constrained, but have remained below freezing throughout the Holocene as evidenced by stable Tunnel sediments. Permafrost in the area is discontinuous and is predominantly present in valley

bottoms, on north-facing slopes, and in poorly drained soils[47]. Permafrost exposed in the Tunnel is syngenetic[48] with ice wedges, segregated ice, reticulate-chaotic ice, and thermokarst cave ice[15]. Ice wedges are up to 3 m wide and some, covering the entire vertical distance of the walls, are more than 5 m tall. Sequences preserved in the Tunnel have radiocarbon ages between ~11,000 and 40,000 years BP[49,50] and are overlain by Holocene deposits[23].

## Data availability

The datasets generated during the study are available from the corresponding author on reasonable request.

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

## Acknowledgements

Funding for T.A.D. was provided by the U.S. Army Engineer Research and Development Program Basic Research (6.1) Program and Department of Defense's Strategic Environmental Research and Development Program (Project RC18-C2–1170). Funding for M.T.M. was provided by NASA grant NNX16H14G.

## Author contributions

T.A.D. and M.T.M. conceived the project; T.A.D. made the Tunnel measurements; M.T. M. performed the modeling; both authors analyzed the data and wrote the manuscript.

## Additional information

**Competing interests:** The authors declare no competing interests.

