## [Peer Review File · Nature Communications]

Reviewers' comments:

Reviewer #1 (Remarks to the Author):

The manuscript by Douglas and Mellon combines observations and theory to better constrain the magnitude of sublimation on both Earth and Mars. The study is based on observations that were made in a tunnel in the permafrost, thanks to the fact that the tunnel was carved 52 years ago, providing a timescale sufficiently large to be more significant than lab based studies. Their conclusion has strong impact on terrestrial studies on permafrost as well as studies on Mars geology. They observe sublimation rates 4 times higher than predicted.

Overall, I think the paper is well done, the sections with terrain data and measurements are well detailed and the implications well written. I have only minor comments before publications. The paper will be a nice contribution to this topic for multiple communities (Geomorphology, Planetary science, Environmental Science).

Minor comments:

- End of page 8: The effective diffusion coefficient can be scaled to a value of: $9.9 \cdot 10^{-4} \text{ m}^2/\text{s}$. There is a difference of two orders of magnitude compared to Earth ($9 \cdot 10^{-6} \text{ m}^2/\text{s}$). First, this difference should be better highlighted. Second, what is the main reason of this difference? Is this more P or T? Or both? Or other parameters? Is it at fixed tortuosity. I miss some explanations here.

- Middle of page 9: It is mentioned that studies were inconsistent pointing to similar values and up to a factor 4x. Is this really so inconsistent? A factor 4 is not two orders of magnitude different and so it is not that meaningful. Which also lead to the question: What are the implications of the finding of your study, in other words, what differences a factor 4 would make in the understanding of landscapes or processes on Earth or Mars? For instance, given the poor knowledge on tortuosity of Mars ground, would the factor of 4 be so significant compared to the parameters controlling the soil properties?

- End of page 6, it is written that equation 3 can be fit to the observations on Figure 4. It would be useful to actually show this curve on the plot (for instance by a thin dashed line).

Reviewer #2 (Remarks to the Author):

Review: " Sublimation of terrestrial permafrost and the implications for ice-loss processes on Mars" by T. A. Douglas and M. T. Mellon.

This manuscript claims the first in situ measurement of the diffusion coefficient of ice-free loess and argues for its relevance to interpretation of features on Mars. The justification for the exclusive nature of the measurement is the unique permafrost tunnel environment, in which sublimation is the dominant erosional process.

The manuscript is well written, of appropriate length, etc. (the only quibble is the repeated use of the word relic where relict is intended).

The highlighted findings are (1) a measured diffusion constant of $.0905 \text{ cm}^2/\text{s}$ compared to previous measurements of $.0735 \text{ cm}^2/\text{s}$, and (2) a sublimation rate 4x larger than predicted by accepted theory, and (3) implications for interpretation of martian geology. Taking these separately:

1) The 23% discrepancy of D relative to prior laboratory measurements is relatively small

compared to the natural variation in properties. Since the measurement is in situ, this might still be worthy of reporting if quantitatively correct but, as the authors note in equation 1, the measured value is the product of $D \Delta N$, where ΔN is the difference in water vapor density at the ice surface and in the tunnel. As a result, referring to steam tables, the same result would be achieved if D were identical to the lab measurement and the ice surface were at -2°C instead of the assumed -4.1°C (i.e. in equilibrium with the air conditioned tunnel air). As stated in the manuscript, the Fairbanks area has a mean annual temperature of -3.4°C as per a cited 2001 study, already nearly a degree warmer than assumed. Considering the influence of climate change since 2001, as well as the fact that mean air temperature and mean ground temperature aren't necessarily the same, -2°C is a plausible ice surface temperature. Since evidence that the wall temperature is in equilibrium with the air conditioned tunnel air was not provided in the manuscript, the new result would have to be considered as within the error bar of the old.

2) A similar argument applies to the anomalous 4x faster sublimation rate than predicted by theory. In that case, the theoretical rate goes as $\Delta\rho/\rho$ (or, since it is nearly isothermal, $\Delta p/p$ where p is partial water vapor pressure). Since RH is maintained at 91% within the tunnel, the measured sublimation rate turns out to be consistent with an ice surface that is only slightly warmer than the air conditioned air in the tunnel. To give a specific example; saturation vapor pressure (SVP) at -4°C is ~ 440 Pa, so at 91% RH, Δp will be ~ 40 Pa and p will be 400 Pa. If the ice surface is at -1°C (SVP ~ 560 Pa), Δp will be 160 Pa, 4x the assumed value. As above, lacking quantitative measurement of the ice surface temperature, the measured rate would seem to fall within the error bars. Thermodiffusion may also contribute to a larger rate if the wall is warmer than the surrounding air.

3) With respect to realistic diffusion constants for Mars, the authors failed to cite the extensive work of Hudson and others (Hudson et al JGR 112 E-5-16 2007; Hudson & Aharonson JGR 113, E09008, 2008 and references therein), which provide more insight than the work here on both the physics of diffusion under martian conditions and the range of expected value.

Reviewer #1 (Remarks to the Author):

The manuscript by Douglas and Mellon combines observations and theory to better constrain the magnitude of sublimation on both Earth and Mars. The study is based on observations that were made in a tunnel in the permafrost, thanks to the fact that the tunnel was carved 52 years ago, providing a timescale sufficiently large to be more significant than lab based studies. Their conclusion has strong impact on terrestrial studies on permafrost as well as studies on Mars geology. They observe sublimation rates 4 times higher than predicted.

Overall, I think the paper is well done, the sections with terrain data and measurements are well detailed and the implications well written. I have only minor comments before publications. The paper will be a nice contribution to this topic for multiple communities (Geomorphology, Planetary science, Environmental Science).

We thank this Reviewer for their time and constructive comments.

Minor comments:

- End of page 8: The effective diffusion coefficient can be scaled to a value of: $9.9 \times 10^{-4} \text{ m}^2/\text{s}$. There is a difference of two orders of magnitude compared to Earth ($9 \times 10^{-6} \text{ m}^2/\text{s}$). First, this difference should be better highlighted. Second, what is the main reason of this difference? Is this more P or T? Or both? Or other parameters? Is it at fixed tortuosity. I miss some explanations here.

Pressure is the main factor, followed by temperature and gas species. The first two sentences in this paragraph state:

“Based on the kinetic theory of gases, the binary diffusion coefficient scales as $T^{3/2}$ and P^{-1} , as noted above. On Mars, the mean temperature is 205K and atmospheric pressure is 600 Pa.”

To address this comment we clarified the difference between Earth and Mars and including gas species and modified the remainder of the paragraph to read:

“Based on the kinetic theory of gases, the binary diffusion coefficient scales as $T^{3/2}$ and P^{-1} , as noted above. On Mars, the mean temperature is -68°C (205K) and atmospheric pressure is 600 Pa, such that the diffusion coefficient for this same soil would be $\sim 109\times$ larger. However, the difference in diffusion through air (Earth atmosphere) and carbon dioxide (Mars atmosphere)²⁷ reduces this scaling by a factor of 1.65 to give a Mars effective diffusion coefficient of $5.99 \times 10^{-4} \text{ m}^2/\text{s}$, $\sim 66\times$ larger. This value is similar to laboratory measurements conducted with glass spheres conducted at Mars⁴¹. This value is also 4 to 5 times larger than has been utilized previously in Mars theoretical studies^{25,26,27,42,3}, which can be mainly attributed to the high porosity and low tortuosity structure of this analog to martian soil.”

- Middle of page 9: It is mentioned that studies were inconsistent pointing to similar values and up to a factor 4x. Is this really so inconsistent? A factor 4 is not two orders of magnitude different and so it is not that meaningful. Which also lead to the question: What are the implications of the finding of your study, in other words, what differences a factor 4 would make in the understanding of landscapes or processes on Earth or Mars? For instance, given the poor knowledge on tortuosity of Mars ground, would the factor of 4 be so significant compared to the parameters controlling the soil properties?

This factor of 4 applies to exposed ice, so tortuosity and other soil parameters do not apply.

To clarify this and directly address this Reviewer’s comment we added a brief list of example relevant applications of the 4x difference:

“On Mars, this difference is important when linking sublimation to the timing of naturally occurring secular or cyclic changes. For example: i) summertime sublimation from the martian polar caps, which is thought to be linked to the global atmospheric humidity⁴⁴; ii) the rate of evolution

of non-polar ice exposures⁷; iii) determining the concentration of soil in ice exposed by recent impacts³³; and iv) the lifetime and instability of transient liquid water^{30, 31, 32}.”

- End of page 6, it is written that equation 3 can be fit to the observations on Figure 4. It would be useful to actually show this curve on the plot (for instance by a thin dashed line).

We have added the curve to an updated Figure 3. This Figure is also presented here:

The caption for Figure 3 has been edited to address this as follows:

“**Figure 4. The 52.5 year record of sublimation measurements from the tunnel.** Upper panel: ice cemented loess yields a log normal relationship with an r^2 of 0.90. **This fit is denoted by the dashed line.** Vertical lines with horizontal bars represent plus and minus one standard deviation of the value for those measurements.”

Reviewer #2 (Remarks to the Author):

Review:” Sublimation of terrestrial permafrost and the implications for ice-loss processes on Mars” by T. A. Douglas and M. T. Mellon.

This manuscript claims the first in situ measurement of the diffusion coefficient of ice-free loess and argues for its relevance to interpretation of features on Mars. The justification for the exclusive nature of the measurement is the unique permafrost tunnel environment, in which sublimation is the dominant erosional process.

The manuscript is well written, of appropriate length, etc. (the only quibble is the repeated use of the word relic where relict is intended).

Thank you for identifying this spelling error. We apologize. This has been fixed in both locations in the manuscript in the section titled “Ramifications for sublimation on Mars and Earth,” fifth paragraph: “Relict ice (unstable ice left over a past climate state) may persist if a sufficient delay of ice-loss follows the most recent period of stability. Rapid diffusion of sublimated water vapor decreases the potential for relict ice to occur.”

The highlighted findings are (1) a measured diffusion constant of $.0905 \text{ cm}^2/\text{s}$ compared to previous measurements of $.0735 \text{ cm}^2/\text{s}$, and (2) a sublimation rate 4x larger than predicted by accepted theory, and (3) implications for interpretation of martian geology. Taking these separately:

1) The 23% discrepancy of D relative to prior laboratory measurements is relatively small compared to the natural variation in properties. Since the measurement is in situ, this might still be worthy of reporting if quantitatively correct but, as the authors note in equation 1, the measured value is the product of $D \Delta N$, where ΔN is the difference in water vapor density at the ice surface and in the tunnel. As a result, referring to steam tables, the same result would be achieved if D were identical to the lab measurement and the ice surface were at -2°C instead of the assumed -4.1°C (i.e. in equilibrium with the air conditioned tunnel air). As stated in the manuscript, the Fairbanks area has a mean annual temperature of -3.4°C as per a cited 2001 study, already nearly a degree warmer than assumed. Considering the influence of climate change since 2001, as well as the fact that mean air temperature and mean ground temperature aren't necessarily the same, -2°C is a plausible ice surface temperature. Since evidence that the wall temperature is in equilibrium with the air conditioned tunnel air was not provided in the manuscript, the new result would have to be considered as within the error bar of the old.

The Reviewer makes a good point here- the temperature information for the Fairbanks area, the ambient air inside the Permafrost Tunnel, and at the wall surface in the Tunnel are all potential variables to consider. We have addressed this in detail here and have clarified the manuscript text as well.

The mean annual temperature in the Fairbanks area, as per Jorgenson, M, *et al.* Permafrost degradation and ecological changes associated with a warming climate in central Alaska. *Clim. Change* 48(4), 551–579 (2001),” is -3.3°C . Note we had -3.4°C in the manuscript and have updated that to -3.3°C . The mean annual temperature in Jorgenson et al., 2001 is for the Tanana Flats region which is about 15 kilometers south of the Permafrost Tunnel. As such, it is a relevant general comparison for the Fairbanks area and thus the Permafrost Tunnel site.

While the quantitative scaling the reviewer uses at -2°C is not correct, the point is taken. We have added a discussion of the uncertainty associated with the tunnel wall temperature, the factors that influence it, and the sensitivity of our findings to the wall temperature value.

The key text additions are as follows. In the subsection “**Vapor diffusion and lag formation**”, third paragraph, we have added:

“The temperature history of the wall-ice surface (the wedge-ice surface or the interface between dry loess and ice-rich loess) has not been measured. We expect the bulk permafrost at the depth of the tunnel to reflect the local mean surface temperature over the past 30 to 100 years, estimated to be in the range of -5 to -3°C ^{19, 20, 29}. However, the active cooling of air within the tunnel will slowly effect the wall surfaces such that: i) the wall temperatures will gradually warm or cool toward the mean air temperature; ii) large swings in air temperature will be greatly subdued by the large thermal mass of the permafrost, compounded by any insulating dry loess layer; and ii) latent heat of sublimation of wall ice will further cool the walls. As such -4.1°C is a reasonable mean wall temperature, but may deviate slightly, the sensitivity of which is discussed below.”

Further, in the subsection “**Ramifications for sublimation on Mars and Earth**”, we have added a new (now) paragraph 8:

“In both cases of diffusion and exposed-ice sublimation a warmer than assumed wall temperature may help to explain some of these differences. Only $\frac{1}{4}^{\circ}\text{C}$ increase in the mean wall temperature would be needed for the diffusion coefficient to match the laboratory measurements of the disturbed loess. A 1°C increase would be needed for observed sublimation to agree with theory. While these temperature increases are small, they are difficult to envision in light of cooler tunnel air temperature and the cooling effect of sublimation itself. What’s more, no value of wall temperature provides a simultaneous match to both diffusion through loess and sublimation. Given the parameterized nature and heat-convection-analogy of the model of ice sublimation, a simpler explanation is that this model under predicts sublimation.”

2) A similar argument applies to the anomalous 4x faster sublimation rate than predicted by theory. In that case, the theoretical rate goes as $\Delta\rho/\rho$ (or, since it is nearly isothermal, $\Delta p/p$ where p is partial water vapor pressure). Since RH is maintained at 91% within the tunnel, the measured sublimation rate turns out to be consistent with an ice surface that is only slightly warmer than the air conditioned air in the tunnel. To give a specific example; saturation vapor pressure (SVP) at -4°C is ~ 440 Pa, so at 91% RH, Δp will be ~ 40 Pa and p will be 400 Pa. If the ice surface is at -1°C (SVP ~ 560 Pa), Δp will be 160 Pa, 4x the assumed value. As above, lacking quantitative measurement of the ice surface temperature, the measured rate would seem to fall within the error bars. Thermodiffusion may also contribute to a larger rate if the wall is warmer than the surrounding air.

Our response to point #1 and the added/edited text in the manuscript also applies to and addresses this comment.

3) With respect to realistic diffusion constants for Mars, the authors failed to cite the extensive work of Hudson and others (Hudson et al JGR 112 E-5-16 2007; Hudson & Aharonson JGR 113, E09008, 2008 and references therein), which provide more insight than the work here on both the physics of diffusion under martian conditions and the range of expected value.

We added reference to Hudson et al 2007 (now reference #41), who report diffusion coefficients measured for glass spheres at Mars pressures that can be compared to our findings. In which case their results are supportively similar, which we now note.

“Hudson TL, Aharonson O, Schorghofer N, Farmer CB, Hecht MH, Bridges NT. Water vapor diffusion in Mars subsurface environments. *Journal of Geophysical Research: Planets*. 2007 May 1;112(E5).”

However, a precise quantitative comparison is not useful due to various differences, grain size and pore size distributions, packing densities, and atmospheric pressures and gas species, such that this comparison with take a path through theoretical modeling containing a number of assumptions we do not feel confident to make from our measurements. Neither of these papers measured the properties of Fox tunnel loess and so their data is of limited application toward our study.

REVIEWERS' COMMENTS:

Reviewer #2 (Remarks to the Author):

The authors have adequately addressed the concerns raised in the first review. These primarily reflect the sensitive dependence of the conclusions on the wall temperature, which was not directly measured. While I would have preferred to see at least a contemporary measurement, the authors have at least acknowledged the sensitivity and justified their assumptions.

I note a couple of typographical errors that crept into the changes, with the missing word marked by brackets:

- "which is thought [to] be linked"
- "conducted with glass spheres conducted at Mars [P] 41."

REVIEWERS' COMMENTS:

Reviewer #2 (Remarks to the Author):

The authors have adequately addressed the concerns raised in the first review. These primarily reflect the sensitive dependence of the conclusions on the wall temperature, which was not directly measured. While I would have preferred to see at least a contemporary measurement, the authors have at least acknowledged the sensitivity and justified their assumptions.

We thank the Reviewer for these comments.

I note a couple of typographical errors that crept into the changes, with the missing word marked by brackets:

- "which is thought [to] be linked"

This has been changed, as suggested, in the section "Ramifications for sublimation on Mars and Earth" to:

"For example: i) summertime sublimation from the martian polar caps, which is thought to be linked to the global atmospheric humidity⁴⁴; ii) the rate of evolution of non-polar ice exposures⁷; iii) determining the concentration of soil in ice exposed by recent impacts³³; and iv) the lifetime and instability of transient liquid water^{30, 31, 32}."

- "conducted with glass spheres conducted at Mars [P] 41."

This has been changed to:

"This value is similar to laboratory measurements with glass spheres conducted at Mars pressures⁴¹"